# DIMENSIONLESS INSTANCE SEGMENTATION BY LEARNING GRAPH REPRESENTATIONS OF POINT CLOUDS

## ABSTRACT

Point clouds are an increasingly common spatial data modality, being produced by sensors used in robotics and self-driving cars, and as natural intermediate representations of objects in microscopy and other bioimaging domains (e.g., cell locations over time, or filaments, membranes, or organelle boundaries in cryo-electron micrographs or tomograms). However, semantic and instance segmentation of this data remains challenging due to the complex nature of objects in point clouds. Especially in bioimaging domains where objects are often large and can be intersecting or overlapping. Furthermore, methods for operating on point clouds should not be sensitive to the specific orientation or translation of the point cloud, which is often arbitrary. Here, we frame the point cloud instance segmentation problem as a graph learning problem in which we seek to learn a function that accepts the point cloud as an input and outputs a probability distribution over neighbor graphs in which connected components of the graph correspond to individual object instances. We introduce the Dimensionless Instance Segmentation Transformer (DIST), a deep neural network for spatially invariant instance segmentation of point clouds to solve this point cloud-to-graph problem. DIST uses an SO(n) invariant transformer layer architecture to operate on point clouds of arbitrary dimension and outputs, for each pair of points, the probability that an edge exists between them in the instance graph. We then decode the most likely set of instances using a graph cut. We demonstrate the power of DIST for the segmentation of biomolecules in cryo-electron micrographs and tomograms, far surpassing existing methods for membrane and filament segmentation in empirical evaluation. DIST also applies to scene and object understanding, performing competitively on the ScanNetV2 3D instance segmentation challenge. We anticipate that DIST will underpin a new generation of methods for point cloud segmentation in bioimaging and that our general model and approach will provide useful insights for point cloud segmentation methods in other domains. [†]

## 1 INTRODUCTION

Point clouds are a common way to represent objects or scenes in a computer, and are widely used in computer vision, augmented and virtual reality, and imaging. Point clouds of locations are often subsequently processed to semantically classify points - semantic segmentation - or to segment individual objects and instances - instance segmentation (**Figure 1**). Unlike 2D or 3D images, point clouds are disordered, unstructured, and may have noisy point locations, making it difficult to design algorithms or machine learning models to process them. Deep learning methods for processing point clouds have become of increasing interest as more and more point cloud data are being generated by sensors in robotics and as a representation of objects in physics engines, natural images, and bioimaging. Many recent methods have been developed to segment point clouds using deep learning (Lai et al., 2022; Qi et al., 2017; Wang, 2020; Zanjani et al., 2021; Guo et al., 2020; Hong and Pavlic, 2021; Pan et al., 2018; Yuan, 2021), which address the instance, scene, or part segmentation problems using various architectures or training schemes. However, instance segmentation methods require prior information about the number of present instances or assume some fixed number of instances. Furthermore, many methods convert point clouds into pixel- or voxel-grids to process them with

---

[†]Code available at **redacted**.

convolutional layers, or otherwise incorporate point coordinates directly into the network, causing their outputs not to be invariant to rotation and translation of the point cloud.

In cryo-electron microscopy, an increasingly common task is to segment individual filaments, membranes, organelles, or other biological structures in 2D micrographs or 3D tomograms. These objects are often large and can intersect or overlap causing instance segmentation to be difficult even if a semantic segmentation mask is known. Experienced scientists or technicians often spend weeks to months painstakingly manually labeling these datasets for downstream analysis, limiting throughput. Current state-of-the-art methods barely help. For filament instance segmentation, for example, these methods utilize algorithms custom-tailored to curve tracing (Chai et al., 2022) but have such high error rates that scientists still spend days manually correcting annotation if the algorithms work at all (Redemann et al., 2014; Stalling et al., 2005). Faster and more accurate instance segmentation methods are urgently needed to facilitate large-scale analysis of these datasets as imaging technology improves.

To address these problems, we propose the Dimensionless Instance Segmentation Transformer (DIST). DIST is able to perform SO(n) invariant instance segmentation of point clouds using only geometric features. We accomplish this by framing instance segmentation as a graph prediction problem. Given a point cloud as input, DIST outputs a probability distribution over graphs parameterized by the probability, for each pair of points, that those points are neighbors in sub-graphs defining each instance. Instances, therefore, are defined by connected components of the full point cloud graph. With the output of DIST, we are able to find the most likely instance segmentation using a graph cut. This allows us to perform instance segmentation on any number of underlying instances without any built-in restrictions on the maximum number of instances. Furthermore, DIST is invariant to rotation and translations of the point cloud, because it operates on point-point pairwise representations initially defined by the distances between the points. This also makes DIST dimensionless as the distance between points is invariant for a number of dimensions. The DIST layers update edge representations using geometrically inspired operations, axial attention updates over the source and destination nodes, and a triangular multiplicative update, inspired by (Jumper et al., 2021). DIST can, optionally, accept additional, non-spatial, point features incorporated via a traditional transformer layer where the attention updates include a bias term learned from the learned edge features. Empirically, we find that DIST performs incredibly well, improving on current state-of-the-art solution for membrane instance segmentation in 2D micrographs and microtubule (MT) segmentation in 3D tomograms by a large margin (from 0.539 mCov for Amira to 0.955 mCov with DIST). DIST also applies to instance segmentation of other point clouds, outperforming other geometric methods for instance segmentation on ScanNetV2 (Dai et al., 2017) and showing competitive results for current state-of-the-art models. In this work, we make the following contributions:

- We frame instance segmentation as a graph learning problem, where we learn a function that maps point clouds to distributions over neighbor graphs in which instances are connected components.

- We introduce the Dimensionless Instance Segmentation Transformer (DIST) to perform SO(n) invariant inference on the instance neighbor graph using the point cloud as input.

- DIST can operate on point clouds with only geometric features and can, optionally, incorporate additional point features.

- DIST does not require prior knowledge about the number of instances in a point cloud and has no built-in limitations on the number of instances that can be segmented simultaneously.

- Empirical results show that DIST dramatically outperforms previous methods for membrane and microtubule segmentation in cryo-electron microscopy data and that DIST outperforms other geometric methods for instance segmentation in natural 3D scene scans.

## 2   RELATED WORK

Recently, interest in point cloud segmentation methods has increased significantly, partly enabled by benchmarks such as ScanNetV2 (**Appendix Table B**) (Dai et al., 2017). Point cloud segmentation tasks are generally divided into semantic, part, and instance segmentation. Deep learning methods for semantic and part segmentation, such as PointNet++ (Qi et al., 2017) and Point Cloud Transformer (Guo et al., 2020) have achieved significant improvements. Instance segmentation methods have

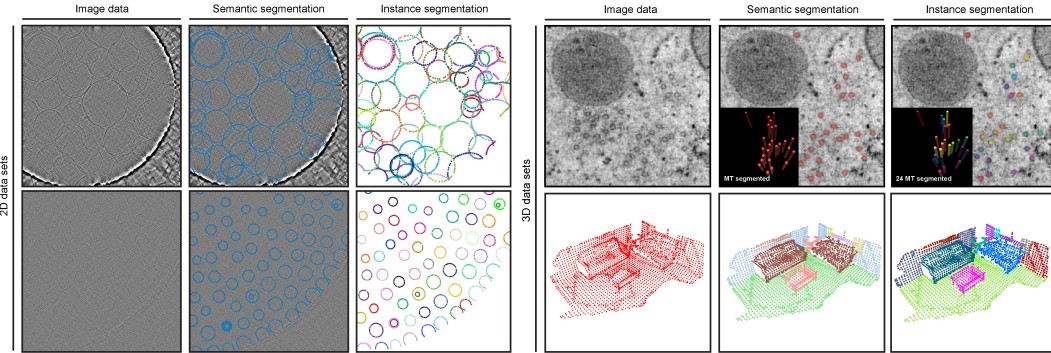

Figure 1: Examples of semantic and instance segmentation in 2D on a cryo-EM micrograph (left), 3D on an electron tomogram (right-top) and scene from ScanNetV2 (right-bottom). Semantic segmentation classifies points into semantic classes. Instance segmentation classifies points into individual object instances, which may be of the same semantic class. Membranes and microtubules, for example, are each a single semantic class, but there can be many membrane or microtubule instances in a single image.

also improved, but these improvements have been slower due to the additional challenge in task formulation. Here, we concentrate on instance segmentation methods as that is the focus of this work. Instance segmentation methods can generally be divided into two groups: proposal-based and proposal-free.

**Proposal-based methods** convert instance segmentation into sub-tasks: 1) find bounding boxes of the objects, and 2) for each detected object, map it back to the input to find an instance segmentation. This requires multi-stage training and pruning incorrect object predictions. Methods such as 3D-SIS (Hou et al., 2018), LiDARSeg (Zhang et al., 2020), and 3D-MPA (Engelmann et al., 2020) use this approach. Because these methods rely on an initial object detection pass with objects defined as bounding boxes, they are not appropriate for instance segmentation when objects cannot be discretely separated into bounding boxes or are otherwise poorly represented by regions.

**Proposal-free methods** generally approach instance segmentation as a clustering step after semantic segmentation. PartNet (Mo et al., 2018) and PointGroup (Jiang et al., 2020b) achieve this with discriminative feature learning and point grouping. However, these methods often struggle to correctly distinguish instance boundaries.

A handful of **graph-based methods** have been considered for operating on point clouds. In these methods, point clouds are generally converted into a k-nearest neighbors graph before being processed with a graph convolutional neural network (GCN; (Guo et al., 2021)). For example, (Simonovsky and Komodakis, 2017) use GCNs for object classification from Sydney Urban Objects dataset and chemical structures where each atom was represented as a node. This was extended by (Pan et al., 2018) to do part and rotated 3D object classification on the PartNet dataset using dynamic GCN. However, there are only two graph-based methods that have been proposed for instance segmentation, to our knowledge, OTOC (Liu et al., 2021) and SegGroup (Tao et al., 2022). These methods output node features that are used for instance segmentation similar to proposal-free methods. In contrast, our method outputs a graph that defines instances directly. It was achieved via a triangular multiplication layer (Jumper et al., 2021) that allowed DIST to learn a geometrical representation of instances. Jumper et. al., use triangular multiplication to update their pair representations, which correspond to each pair of columns in their multiple sequence alignment. This was used to learn triangle inequality in their MSA representation. In our study, we observed that this approach also allows for defining a distance-inspired update for our edge/pairwise point representations. Jumper et. al., apply these updates sequentially to rows and columns whereas we apply them independently to each and then accumulate those updates. We also include row and column axial attention updates which improved our ability to resolve object instances that were close to each other. Furthermore, our method does not require semantic input features, unlike OTOC and SegGroup. DIST also outperforms OTOC and SegGroup by a considerable margin on the ScanNetV2 instance segmentation dataset.

## 3 METHOD

Instance segmentation with DIST is divided into three parts (**Figure 2**): **1)** the raw point cloud is converted into a pairwise representation based on distances between the points, **2)** the edge representations are fed through the DIST model to produce an output matrix containing the probability, for each pair of points, that there is an edge between them in the instance graph, and **3)** the maximum likelihood graph defined by the edge probabilities is found using a graph cut algorithm. This finally predicted graph defines the instance segmentation. The DIST model is trained to predict the ground truth edges in labeled instance graphs using standard neural network training techniques, back-propagation and stochastic gradient descent.

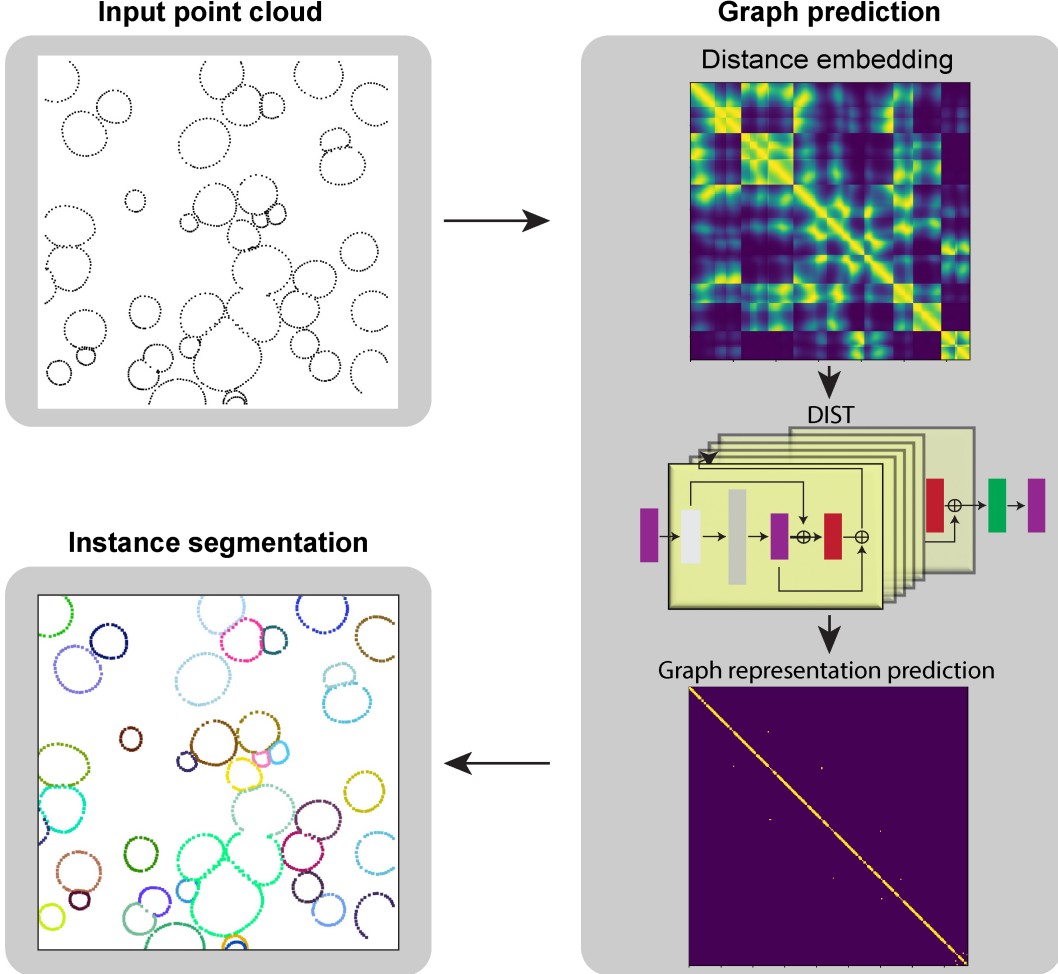

Figure 2: Instance segmentation with DIST. The input pairwise feature representations are defined from the Euclidean distance between points in the input point cloud. The DIST model learns to refine these representations and then outputs edge probabilities. These edge probabilities are used to find the most likely graph, the connected components of which define individual instances.

### 3.1 PAIRWISE-FEATURE INPUT EMBEDDING

The DIST model operates on pairwise features. That is, for each pair of points in the point cloud, DIST learns a vector representation. We refer to these as pairwise embeddings or edge embeddings. In order to ensure that our representations are invariant to the translation and rotation of the point cloud, we initialize the edge features using the corresponding distance between each pair of nodes. Because we think that nearby points are more important than distant points within the point cloud, we define these features using a scaled exponential of the negative squared distance. More formally,

given a point cloud represented as a set of points $p_i$ where $i = 1, \ldots, n$, we initialize a graph $G$ with a featureless node for each $p_i$, and edges $e_{i,j}$ connecting all nodes, with weights $e_{i,j} = exp(\frac{-d_{i,j}^2}{s^2*2})$, where $d_{i,j}$ is the euclidean distance between $p_i$ and $p_j$, and $s$ is sigma denoted as fixed scaling factor for the normalized point cloud. This weighting approach allows us to embed geometrical information about all points directly from the Euclidean distance. These weights are then multiplied with a learned $d-$dimensional embedding vector to define the initial edge representations.

## 3.2 Geometric Transformer Layer

Starting from the initial edge representations, we apply several steps of axial multi-head attention (MHA) and triangular multiplicative updates (**Appendix Figure C-B**). Each overall step is referred to as a "DIST layer" (**Appendix Figure C-C**), and the edge update that occurs within each DIST layer (composed of simultaneous MHA and triangular multiplicative updates) is referred to as the "Edge update" module (**Appendix Figure C-D**). In the following description, we describe actions on rows/columns of the graph representation of $G$, which corresponds to the set of incoming or outgoing edges of $G$.

**MHA module** was adapted from the original paper (Vaswani et al., 2017). MHA was chosen to increase stability and captures the relationships between node/edges with every node/edge in the point cloud. It achieved it by performing axial attention updates over rows or columns of the graph. In axial updates, we switch between interpreting the rows or columns as the long axis of the tensor and apply typical multi-head attention. This corresponds to attending over edges outgoing from a node (row attention) or incoming to a node (column attention). This is followed by a fully connected layer with residual connections from the row and column axial attention operations to generate the output edge features.

**Triangular multiplicative update** was adapted from (Jumper et al., 2021), with a detailed design depicted in **Appendix Figure C-E** and pseudo-code shown in **Appendix D**. We chose to perform this operation to allow DIST to directly learn the relationship between each node. Triangular update in Jumper work was performed by a sequential operation over rows and columns, which was essential to learn MSA representation. In our approach, we perform the triangular operations at the same time for rows and columns which allowed us to better learn of geometrical features of a point cloud. The triangular multiplicative update could achieve it by utilizing point distances encoding their geometrical features from which triangular update could learn instance representation. The triangular update achieves it by taking an input edge feature embedding and performing Einstein summation over row or column features, followed by a linear layer with gating to obtain an update for the edge feature embedding.

At each DIST layer, the output from both the MHA and triangular multiplicative updates are added to the incoming edge features, and the resulting updated edge features are fed through an activation function before being used as the input to the next DIST layer. Finally, after the last DIST layer, the resulting edges in the graph are fed through a single linear layer to obtain a probability for that edge.

## 3.3 Graph Cut Instance Segmentation

The final DIST graph representation contains the predicted probability for each edge in the instance graph of the point cloud, which can be used to obtain an instance segmentation. A simple segmentation can be achieved by thresholding the probabilities and examining the resulting graph.

For some problems, we may have additional information about the graph structure. For example, for filament and membrane segmentation, we know the graph is defined as a linear chain in which a node can have at most 2 neighbors. In this case, we adopt a greedy algorithm for graph inference (**Appendix E**). First, we build a hash map for each node in the point cloud containing node ID $P_i$ and edge probability $p_{i,j}$. Next, for each new instance, we searched the hash map for the initial node. This allows us in the final step to iteratively find up to two edges with the highest probability for all connected nodes. This process is continued until no new edges can be recognized for a given instance. We use this approach when segmenting membranes and microtubules which follow this chain structure.

### 3.4 IMPLEMENTATION DETAILS

#### 3.4.1 MODEL HYPERPARAMETERS

In all experiments, we use a DIST model with 6 layers, 8 attention heads, and a hidden dimension of 128. We lightly tuned the value of sigma, finding that both biological datasets had a value of 2, and for ScanNetV2 a value of 0.05 gave the best results on the training set. All DIST models were trained with binary cross entropy loss using the ADAM optimizer with learning rate $10^{-5}$ and no weight decay on a single NVIDIA A100 GPU. DIST model training was stopped when the validation loss did not improve after 50 consecutive epochs.

#### 3.4.2 POINT CLOUD PRE-PROCESSING

There are two major challenges in point cloud instance segmentation: **1)** uneven point spacing within and between datasets, which makes learning difficult to learn generalizable local features, and **2)** the size of the point cloud, which is often too large to process due to memory constraints, because the RAM usage of DIST scales cubically (axial attention requires quadratic attention for each point) with the number of points, making large point clouds require far more RAM than is available on current GPUs.

**Point cloud re-scaling.** We tackle the uneven sampling resolution problem by re-scaling the point cloud. We scale the point clouds for biological data - where point clouds are derived from image data - by using the physical pixel size and normalizing the data such that distances between points are scaled in Angstroms. In the case of ScanNetV2 datasets, no additional point cloud normalization was done.

**Cropping and stitching.** Next, we tackle the memory cost of computing. In the case of the DIST model, the size of the point cloud poses a challenge due to cubic complexity. This makes it impossible to process very large scenes due to GPU memory constraints. Although many methods for improving memory efficiency were demonstrated (Kyzirakos et al., 2016; Hui et al., 2021; Liu et al., 2019), all of them achieve it at the cost of computation speed, down-sampling, or losing semantic or geometrical information. We define the instance segmentation problem as the segmentation of geometrically similar objects. Therefore, we expect most of the information to be present within local regions. With this in mind, we reduced the size of the point clouds by cropping (**Appendix Figure A**). During training, we select crops at random. For inference, we split the point cloud into tiled regions and then use overlap between the regions to stitch the graph across region boundaries. For each dataset, we selected the maximum crop size that would fit into GPU RAM.

## 4 EXPERIMENTS

**Datasets.** Because our primary motivation is to perform instance segmentation on biological structures in electron microscopy (EM) data, we compare the performance of DIST to the current state-of-the-art algorithms for instance segmentation of filament-like structures on real-world EM data. We evaluate DIST on a membrane segmentation task in 2D cryo-EM micrographs and a microtubule (MT) segmentation task in 3D plastic section tomograms. For both datasets, ground truth labels are derived from manual annotation by an experienced microscopist. The datasets contain 76 micrographs of membranes (2D scenes; with 500-2'500 points each) and 48 MT tomograms (3D scenes; with 10'000-50'000 points each). We split these into a train, validation, and test sets with an 80/10/10 split ratio. We also evaluate DIST for instance segmentation of point clouds containing natural objects and benchmark it against other instance segmentation methods on the ScanNetV2 3D instance segmentation challenge (Dai et al., 2017). For ScanNetV2, we train on the standard training set and report results on the validation set.

**Ground truth generation.** In order to train and evaluate the DIST model we need to build an instance graph representation of the point cloud. The graph representation is a 2D matrix with nodes ($P_i, j$; matrix diagonal) representing each individual coordinate point, and the edge represent spatial connectivity between two nodes $i$ and $j$, where $i$ is the matrix row, and $j$ is matrix column. The graph representation for filament-like structures (membranes and microtubules) is constructed from ground truth annotations in which each individual instance is an ordered list of nodes. Knowing the

order, the edge matrix is defined as 1 for each $P_{i,j}$ if $i$ is in the same instance as $j$ and $j$ is a neighbor of $i$ in the ordered list. This approach imposes restrictions where each $i$ can have only up to two edges. In contrast, the ScanNetV2 dataset consists of semantically labeled point clouds obtained from Lidar-scanned scenes. In this case, we construct the graph by labeling as 1 each $P_{i,j}$, if $i$ is in the same instance as $j$ and if the distance from $j$ to $i$ is less than the average $k$NN distance between all points. In both cases, the instances are connected sub-graphs of the full point cloud graph.

**Graph representation evaluation.** We performed experiments on 2D and 3D point clouds manually segmented by experienced users. The evaluation metric used for the graph prediction is intersection over union (mIoU) averaged over all cropped graphs. Due to the lack of a comparable real or synthetic benchmark dataset to which we could compare our DIST graph prediction, we defined here our baseline independently. The baseline was defined as a point cloud graph representation generated from point distances. This was achieved by computing the distance between each pair of nodes in a point cloud.

**Instance segmentation evaluation.** To benchmark instance segmentation performance for membrane and microtubules datasets we measured mean class coverage (mCov; (Jiang et al., 2020a)). This metric measures IoU between the ground truth label and its matching predictions. Additionally, for this dataset, we also generate our baseline using state-of-the-art segmentation software ZiB Amira (Stalling et al., 2005) and multi-curve fitting (MCF; (Chai et al., 2022)). Both of this software are nowadays wildly used for MT segmentation tasks. For both of the methods, we used the 'standard' setting. In the case of ZiB Amira, the setting was heavily tuned for the MT dataset and is suggested to use by the authors. In the case of MCF, the author optimized the 'standard' setting for filament-like structures. MT and Mem datasets are filament-like structures, and therefore using a setting tuned by the author and adjusting only the pixel size value in the author's opinion should yield the best results. Keeping the same with the ScanNetV2 datasets were evaluated using average precision (AP50) denotes the scores with an IoU threshold of 50% and were compared with InsConv, SegGroup and OTOC models. The ScanNetV2 evaluation was then compared against the ScanNetV2 semantic instance segmentation benchmark. The mAP50 score in this benchmark denotes an average AP50 of every predicted instance without taking into account semantic labels which DIST did not produce. Our DIST model predicts only instances, not semantic labels. Class names were obtained post-instance prediction by comparing the GT with predicted instances. then the class name was selected for each instance based on the highest metric value and it was used to demonstrate the AP50 score per class.

## 4.1 MEMBRANE AND MICROTUBULE DATASETS

The metrics of the point clouds were shown in **Table 1**. For the inference of the graph representation, the output of the DIST was thresholded before calculating mIoU. The threshold was selected by manually picking the best value based on a sub-sample of 10 datasets that were not used for the evaluation. The proposed DIST achieved a great leap in performance comparing mIoU of 30% and 113% compared to our baseline **Figure 3**.

| Model | mIoU | | | AUPR | | |
|---|---|---|---|---|---|---|
| | Mem | MTs | ScanNetV2 | Mem | MTs | ScanNetV2 |
| DIST | **0.934** | **0.916** | **0.954** | **0.967** | **0.994** | **0.968** |
| Baseline (point distances) | 0.718 | 0.430 | 0.144 | 0.861 | 0.923 | 0.942 |

Table 1: Graph prediction comparison biological and synthetic datasets.

The evaluation of the graph instance inference model is shown in **Table 2** and **Table 3**. Due to the uniqueness of the EM data, we also compared our DIST framework with the current state-of-the-art method used for membrane and MT automatic segmentation. On the 2D dataset, we compared DIST to two methods: MCF (Chai et al., 2022) and distance clustering (DC). MCF was recently demonstrated as a workflow for instance segmentation based on iterative fitting of points to spline based on multiple hyperparameters that have to be tuned for each dataset. The DC method on the other hand relies on building instances by clustering points based on $k$NN distances. We compared our 3D MT dataset to MCF and currently used Amira software which produces state-of-the-art performance on MT segmentation task (Stalling et al., 2005). This software deals with MT segmentation by iterative searching of neighboring points within a restricted cone shape area. The result overwhelmingly

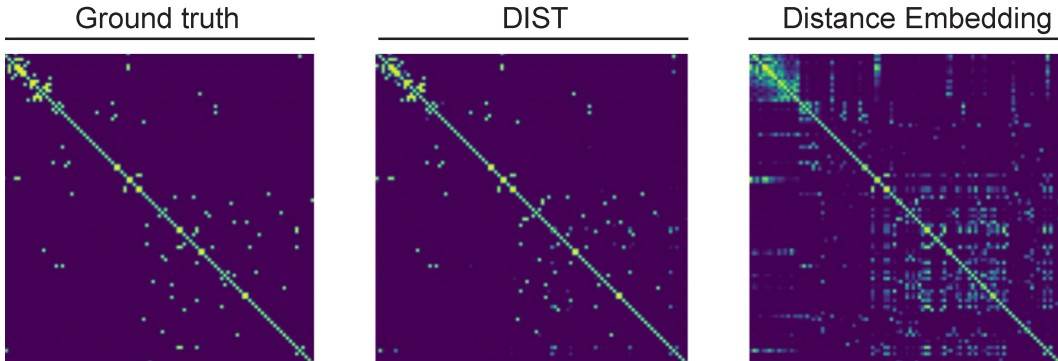

Figure 3: Example of graph prediction.

shows that DIST makes a huge improvement over currently used methods. DIST achieved the best results for both membrane and MT segmentation. **Appendix Figure F** shows further segmentation examples provided by DIST, DC, MCF and Amira. Moreover, MT data in comparison to current state-of-the-art software, Amira, shows remarkably good results (**Table 2** and **Appendix Figure F**).

## 4.2 SCANNETV2 DATASET

We also evaluated our DIST model on the synthetic dataset to see how our model performs in comparison with other published methods. First, we evaluated graph prediction performance for ScanNetV2 datasets similar to what we did for biological data. We observed that our DIST method shows significant performance in predicting graph representation over our baseline by 6.62x. Next, we moved to evaluate our DIST model to see how it performs in comparison with other published methods. For this, we compared our model with SegGroup (Tao et al., 2022), and OTOC (Liu et al., 2021). Bot of this model using point cloud geometry and color features to learn graph representation of the given scene per voxel or point cloud grouping. Additionally, we also compared DIST with the Ins-Conv method (Liu et al., 2022) that was published on scan-net.org 3D semantic-instance segmentation benchmark and was noted not to use color for the prediction of instances, which was similar to our approach. Comparing the DIST model, we observed that our novel approach achieves a significant leap in performance while at the same time using only geometrical information of the point cloud **Table 3** and **Figure F**. We also demonstrated that DIST achieved competitive results with the current state-of-the-art model evaluated on the ScanNetV2 3D instance segmentation task while only using the point distances feature **Appendix Table G**.

| Method | mCov | |
|---|---|---|
| | Mem | MTs |
| DIST | **0.913** | **0.955** |
| DC | 0.253 | 0.214 |
| MCF (Chai et al., 2022) | 0.135 | 0.000 |
| Amira (Stalling et al., 2005) | - | 0.539 |

Table 2: Instance segmentation comparison for the membrane and MT dataset.

## 5 ABLATION STUDY

We conducted an ablation study on the same membrane and MT datasets to evaluate how parts of the edge update module improve DIST performance **Appendix Table H**. We compared the DIST framework with all edge update modules with DIST with only self-attention of the triangular update mechanism turned on. The results show that both transformer and triangular multiplicative updates contribute to the final result. We found that DIST with a full edge update module produced the best result for both datasets. We could also observe that the triangular update added a substantial improvement to DIST in comparison to the transformer update module.

| Model | AP50 | Bathtub | Bed | Bookshelf | Cabinet | Chair | Counter | Curtain | Desk | Door | Otehrfur | Picture | Refrig. | Showe Cur. | Sink | Sofa | Table | Toilet | Window |
|---|---|---|---|---|---|---|---|---|---|---|---|---|---|---|---|---|---|---|---|
| DIST | **0.67** | 0.51 | 0.65 | **0.80** | **0.65** | 0.73 | **0.56** | 0.58 | **0.61** | **0.58** | **0.65** | **0.56** | **0.77** | 0.67 | **0.67** | 0.76 | **0.75** | 0.82 | **0.65** |
| InsConv | 0.66 | **1.00** | 0.76 | 0.67 | 0.58 | **0.86** | 0.32 | 0.66 | 0.48 | 0.47 | 0.55 | 0.43 | 0.65 | **1.00** | 0.66 | 0.74 | 0.59 | 0.94 | 0.47 |
| SegGroup | 0.45 | 0.67 | **0.77** | 0.19 | 0.32 | 0.66 | 0.00 | 0.41 | 0.13 | 0.38 | 0.27 | 0.22 | 0.48 | 0.71 | 0.45 | 0.63 | 0.51 | **1.00** | 0.22 |
| OTOC | 0.53 | 0.67 | 0.72 | 0.78 | 0.40 | 0.68 | 0.00 | **0.67** | 0.14 | 0.39 | 0.37 | 0.54 | 0.36 | 0.64 | 0.56 | **0.77** | 0.59 | 1.00 | 0.25 |

Table 3: Instance segmentation comparison for the ScanNetV2 dataset.

## 6 CONCLUSION

We propose DIST, a neural network architecture for instance segmentation of point clouds. By using geometric features to learn SO(n) invariant graph representations, we are able to achieve state-of-the-art instance segmentation of biological structures in electron micrographs and tomograms and demonstrate competitive results on natural scene understanding in ScanNetV2. By framing the instance segmentation problem as a graph prediction problem where the instances are defined by connected sub-graphs, we are able to identify any number of instances without any built-in constraints in our network. Furthermore, we show that using geometrically inspired updates, the triangular update module, was critical for model performance and that this was enhanced by the inclusion of axial attention modules.

We expect that this approach to instance segmentation will transform our ability to understand biological structures in the increasing amounts of structural data being generated by electron microscopy, where biomedical researchers are in need of fast and accurate instance segmentation methods. In the future, we expect that DIST will underpin filament, membrane, and organelle segmentation software. Furthermore, DIST can be applied to other point cloud instance segmentation problems and extends easily to arbitrary dimension point clouds. This would enable it to be applied to object tracking over time in 3D imaging, for example, which can be represented as a 4D space. DIST can also be extended to include node features, allowing additional semantic information for each point to be passed to the network, further increasing performance.

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

## A    POINT CLOUD PRE- AND POST-PROCESSING

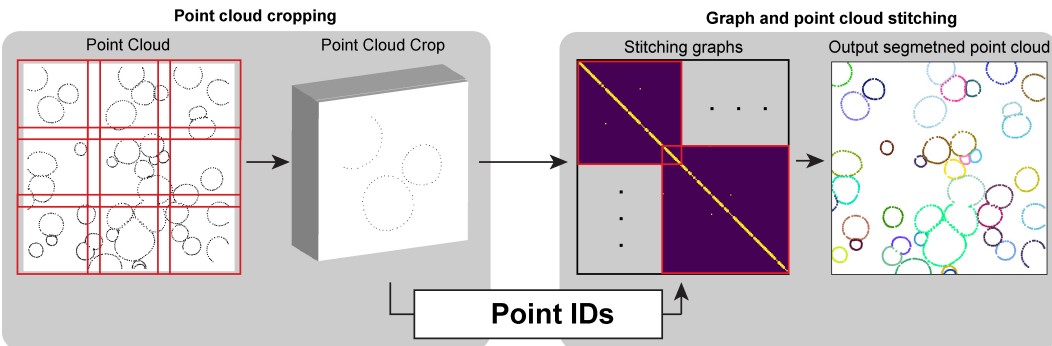

**Appendix Figure 1:** Illustration showing point cloud preprocessing process.

# B  INSTANCE SEGMENTATION METHOD COMPARISON

| Model | Year | mAP50 | Colors | Geometry | 3D |
|---|---|---|---|---|---|
| **DIST (our)** | **2022** | **0.665** | **-** | **+** | **+** |
| Mask3D | 2022 | 0.780 | + | + | + |
| SPFormer | 2022 | 0.770 | + | + | + |
| SoftGroup++ | 2022 | 0.769 | + | + | + |
| SoftGroup | 2022 | 0.761 | + | + | + |
| GraphCut | 2022 | 0.732 | + | + | + |
| DKNet | 2022 | 0.718 | + | + | + |
| SSEC | 2022 | 0.700 | + | + | + |
| HAIS | 2021 | 0.699 | + | + | + |
| SSTNET | 2021 | 0.698 | + | + | + |
| SphereSeg | 2021 | 0.680 | + | + | + |
| Box2Mask | 2022 | 0.677 | + | - | + |
| OccuSeg+instance | 2019 | 0.672 | + | + | + |
| Mask-Group | 2020 | 0.664 | + | + | + |
| **INS-Conv** | **2021** | **0.657** | $-^*$ | **+** | **+** |
| CSC-Pretrained | 2018 | 0.648 | + | + | + |
| PE | 2020 | 0.645 | + | + | + |
| PointGroup | 2019 | 0.636 | + | - | + |
| DD-Unet+Group | 2021 | 0.635 | + | + | + |
| **OTOC** | **2022** | **0.529** | **+** | **+** | **+** |
| Sparse R-CNN | 2020 | 0.515 | + | + | + |
| Occipital-SCS | 2019 | 0.512 | + | + | + |
| 3D-BoNet | 2019 | 0.488 | + | + | + |
| PanopticFusion-inst | 2019 | 0.478 | + | + | + |
| SPG-WSIS | 2022 | 0.470 | + | - | + |
| SALoss-ResNet | 2018 | 0.459 | + | + | + |
| MACS | 2019 | 0.447 | + | + | + |
| **SegGroup** | **2022** | **0.445** | **+** | **-** | **+** |
| 3D-SIS | 2018 | 0.382 | + | + | + |
| Hier3D | 2021 | 0.323 | + | + | + |
| Unet-Backbone | 2018 | 0.319 | + | + | + |
| R-PointNet | 2018 | 0.306 | + | + | + |
| Region-18class | 2022 | 0.284 | + | + | + |
| SemRegionNet | 2021 | 0.250 | + | - | + |
| 3D-BEVIS | 2019 | 0.248 | + | + | + |
| ASIS | 2021 | 0.199 | + | - | + |
| SGPN | 2018 | 0.143 | + | + | + |
| MaskRCNN | 2018 | 0.058 | + | + | + |

$^*$Scan-net.org discrepancy. The original paper of Ins-Conv denotes that colors were used as node features. However, the Scan-net.org benchmark collection indicates that Ins-Conv did not use color values as input data.

**Appendix Table 1:** Instance segmentation comparison for the ScanNetV2 dataset. Data collected from scan-net.org benchmark 3D instance segmentation challenge. Highlighted models indicate networks whose approach does not use colors for feature embedding or use graph representation for predicting instances.

## C  DETAIL MODEL ARCHITECTURE

**Appendix Figure 2:** Detail DIST workflow.
**A)** Distance features embedding for point cloud coordinates. **B)** Pairs between nodes representation denoted as edges on the graph. Triangle multiplicative update is shown. The circles represent nodes with arrows indicating edges used for triangle update. **C)** DIST block. Arrows show information flow. **D)** Update module for edge representation on the graph. Distance embedding features are taken as input and are used for MHA and triangular updates. **E)** Detail structure of the triangular multiplicative update module.

## D  TRIANGULAR MULTIPLICATIVE EDGE FEATURE UPDATE

---

**Algorithm 1** Triangulation algorithm

---

**Require:** $edge_features$

$\quad edgefeatures \leftarrow layerNorm(edgefeatures)$ $\qquad \qquad \triangleright$ Initial normalization of edge features

**Ensure:**

$\quad a = torch.sigmoid(Linear(edgefeatures)) * Linear(edgefeatures)$

$\quad b = torch.sigmoid(Linear(edgefeatures)) * Linear(edgefeatures)$

$\quad$ **if** $axis == 2$ **then**

$\quad \quad k = einsum(biko, bjko- > bijo, a, b)$

$\quad$ **else**

$\quad \quad k = einsum(bkio, bkjo- > bijo, a, b)$

$\quad$ **end if**

$\quad o = torch.sigmoid(Linear(edgefeatures)) * Linear(LayerNorm(k))$

---

# E GREEDY ALGORITHM

---

**Algorithm 2** Greedy algorithm for graph inference

---

**Require:** $C_{patch}^k$ $k \leftarrow P_3^{local}$

**Require:** $G_{patch}^{i,j}$ $i \leftarrow P_i^{local}$ $j \leftarrow P_j^{local}$

**Require:** $IDX_{patch}^k$ $k \leftarrow P_{ID}^{global}$

 **for** $coord, graph, index \leftarrow C_{patch}^k, G_{patch}^{i,j}, IDX_{patch}^k$ **do**       ▷ **Adjacency matrix**
  **for** $i, j \leftarrow graph$ **do**
   $adjacency \leftarrow index_i^k$
   $adjacency \leftarrow coord_i$
   $adjacency \leftarrow index_j^{graph_j >= threshold}$
   $adjacency \leftarrow graph_j^{graph_j >= threshold}$
  **end for**
 **end for**

 $segment = []$                 ▷ **Zero-out greedy algorithm**
 $stop = False$
 $segment_{id} = 0$

 **while** not $stop$ **do**            ▷ **Greedy instance segmenter**
  $ids \leftarrow len(adjacency_{index_j}) >= 0$     ▷ Pick initial point from Adjacency
  $NewSegment \leftarrow ids$

  $growing = False$
  **while** not $growing$ **do**      ▷ Find all nodes associated with initial node
   $size = len(NewSegment)$

   **for** $id \leftarrow ids$ **do**
    *Pick all points associated with the initial point*
    *Check 1: Check if id is not already on the $NewSegment$*
    *Check 2: Check if id is reversible connected $i \leftarrow j$ and $j \leftarrow i$*
    *Check 3: Check if id is not associated to already segmented instance*

    *Add new nodes to $NewSegment$*
   **end for**

   **if** $len(NewSegment) == size$ **then** $growing = True$
   **end if**
  **end while**

  $Sort(NewSegment)$        ▷ **Optional**: Sort point in filament
  $Smooth(NewSegment)$       ▷ **Optional**: Smooth point in filament

  $segment.append(segment_{id} + NewSegment)$
  $prune(NewSegment)$       ▷ Remove segmented nodes from adjacency

  **if** $sum(adjacency) == 0$ **then** $stop = True$   ▷ Stop segmenter when there no more points
  **else**
   $segment_{id}$ += *1*
  **end if**
 **end while**

---

## F    COMPARISON OF DIST INSTANCE SEGMENTATION PERFORMANCE

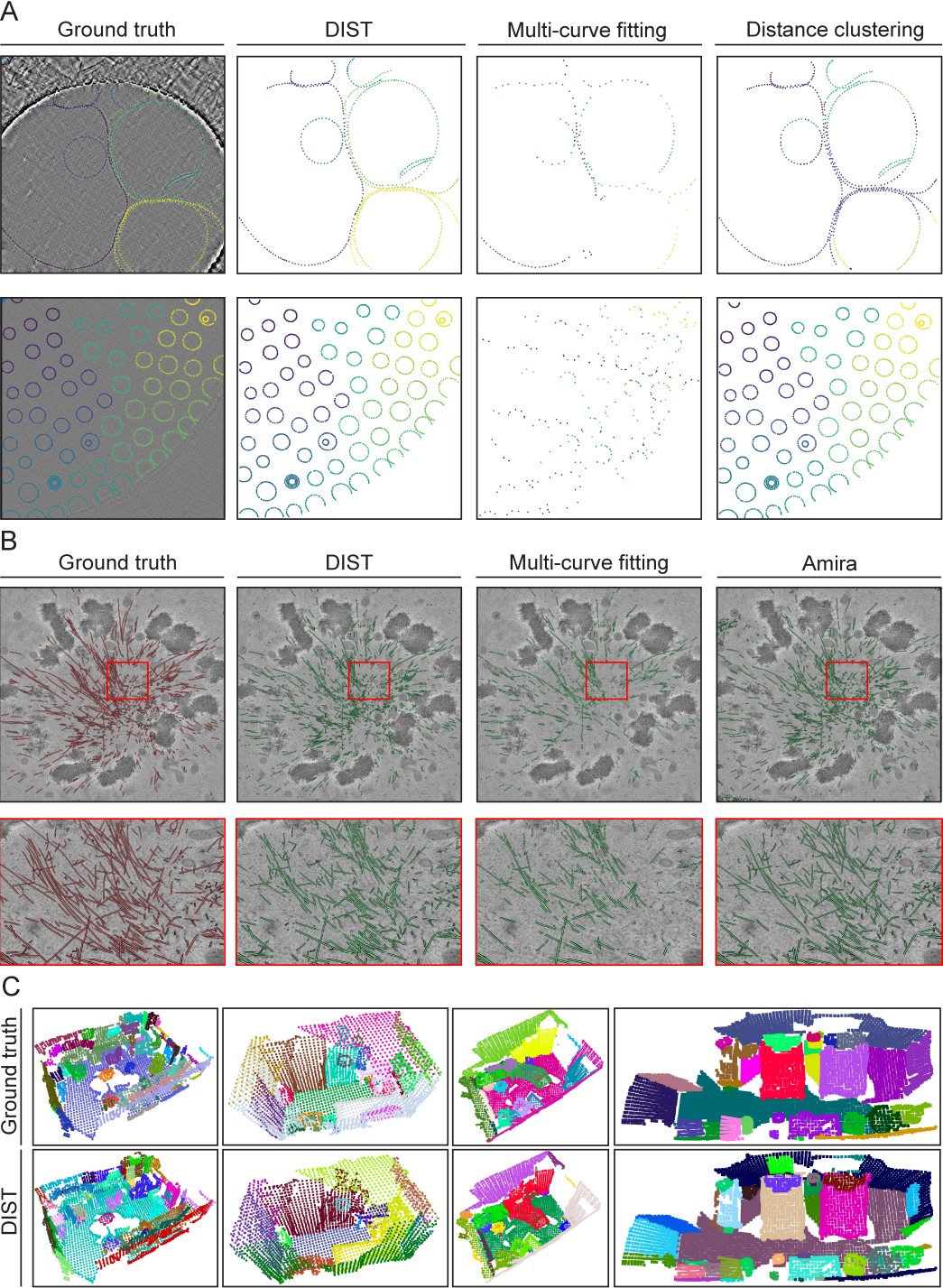

**Appendix Figure 3:** Example of instance segmentation using DIST.
(**A**) Illustration of 2D cryo-EM micrograph with segmented membranes. Instance segmentation was compared with MCF and DC methods. (**B**) Electron tomography reconstruction of mammalian cells with segmented MTs in 3D. Instance segmentation was compared with MCF and Amira methods. (**C**) ScanNetV2 dataset example compared with DIST prediction.

## G  DIST COMPARISON WITH SCANNETV2 STATE-OF-THE-ART MODELS.

| Model | AP50 | Bathtub | Bed | Bookshelf | Cabinet | Chair | Counter | Curtain | Desk | Door | Oehrflr | Picture | Refrig. | Showe Cur. | Sink | Sofa | Table | Toilet | Window |
|---|---|---|---|---|---|---|---|---|---|---|---|---|---|---|---|---|---|---|---|
| DIST (our) | 0.67 | 0.51 | 0.65 | 0.80 | 0.65 | 0.73 | 0.56 | 0.58 | 0.61 | 0.58 | 0.65 | 0.56 | 0.77 | 0.67 | 0.67 | 0.76 | 0.75 | 0.82 | **0.65** |
| **Mask3D** | **0.78** | **1.00** | 0.79 | 0.72 | **0.70** | **0.90** | 0.50 | 0.71 | **0.81** | 0.67 | 0.72 | 0.68 | **0.81** | **1.00** | **0.83** | 0.83 | 0.79 | **1.00** | 0.60 |
| SPFormer | 0.77 | 0.90 | **0.90** | 0.81 | 0.61 | 0.89 | **0.57** | 0.82 | 0.71 | **0.71** | 0.66 | 0.65 | 0.69 | **1.00** | 0.79 | 0.81 | 0.78 | **1.00** | 0.58 |
| SoftGroup++ | 0.77 | **1.00** | 0.80 | **0.94** | 0.68 | 0.87 | 0.21 | **0.87** | 0.67 | 0.57 | **0.76** | **0.70** | 0.81 | **1.00** | 0.65 | **0.90** | **0.79** | **1.00** | 0.63 |

**Appendix Table 2:** Instance segmentation comparison for the ScanNetV2 dataset.

## H  ABLATION STUDY

| Ablation | mIoU | |
|---|---|---|
| | Mem | MTs |
| DIST | 0.934 | **0.916** |
| DIST – with self-attention | 0.863 | 0.776 |
| DIST – with triangular update | **0.985** | 0.904 |

**Appendix Table 3:** DIST ablation study using the Mems and MTs datasets evaluated on graph prediction.

| Ablation | mCov | |
|---|---|---|
| | Mem | MTs |
| DIST | **0.913** | **0.955** |
| DIST – self-attention | 0.641 | 0.441 |
| DIST – triangular update | 0.884 | 0.636 |

**Appendix Table 4:** DIST ablation study using the membrane and MT datasets evaluated on instance segmentation.

