# OpenReview forum: "Dimensionless instance segmentation by learning graph representations of point clouds"
_ICLR.cc/2023/Conference — Submitted to ICLR 2023_

### Official Review · Reviewer_VdTC · 2022-10-24

**Confidence:** 5
**Correctness:** 2
**Technical Novelty And Significance:** 2
**Empirical Novelty And Significance:** 2
**Recommendation:** 3

**Clarity, Quality, Novelty And Reproducibility:**

The reviewer has concerns about the novelty and tech contributions. The developed approach falls short in them given the fact that they merely adopted the existing known techniques including the multi-head attention (MHA) module [1] and triangular multiplicative updates [2] without adding a further major innovation. Additionally, the rational explanation for choosing these components remains untouched, which is likely to confuse the reader and is less useful for understanding the paper. The take-home message seems to just apply the advanced designs in attention models with an efficient way to update edge features and the paper is limited in novelty and tech contributions.

**Strength And Weaknesses:**

The authors tackle an important research problem of point cloud instance segmentation and highlight the importance of accurate instance segmentation to the cryo-electron microscopy tasks, which looks promising for future exploration.
The authors developed some pre-processing techniques to handle the issues of uneven point spacing  (via point cloud re-scaling) and large memory overhead (via cropping and stitching), all of which are useful in practical applications.


There exist some terminologies without further explanation.  For example, the authors referred to their method as a Dimensionless Instance Segmentation Transformer (DIST) but never gave an explanation of the ‘’Dimensionless“ part of the DIST. The authors mentioned SO(n) invariant property for DIST but never formally demonstrated it.

The presentation quality of the paper is not good. For example, Tables 1, 2, 3, and 4 are presented but never get referred to in the main paper. The reviewer can not find Tables 4.1 and 4.2 and Figure 4.1 mentioned multiple times by the authors. All these issues cause significant difficulty to understand the paper. The reviewer tends to believe that this paper is prepared in a rush with poor presentation quality.


**Summary Of The Paper:**

The paper treats instance segmentation as a graph learning problem, via casting a point cloud as a graph, followed by a feature extraction pipeline of applying the multi-head attention (MHA) module [1] and triangular multiplicative updates [2]. The final instance segmentation predictions are obtained by finding the connected components in the graph using a graph cut. Experimental results show that DIST archives competitive results in membrane and microtubule segmentation, and instance segmentation in natural 3D scene scans.

**Summary Of The Review:**

Overall, this paper has a major issue with novelty, tech contribution, and paper presentation. The reviewer tends to give rejection to the current paper due to this reason.

---

> ### Author Response · Authors · 2022-11-17
> **Response to reviewer VdTC**
>
> We are grateful for the reviewer’s comments. We hope that the following response will encourage you to change your opinion about our manuscript.
>
>
> Strength And Weaknesses:
>
> “...never gave an explanation of the "Dimensionless" part of the DIST. The authors mentioned SO(n) invariant property for DIST but never formally demonstrated it.”
>
> We did formally explain that our method operates only on the distances between points. This inherently determines that DIST is invariant to the translation and rotation of the point cloud, and can be applied to points in any dimensional space. This was included in the introduction “Furthermore, DIST is invariant to rotation and translations of the point cloud because it operates on point-point pairwise representations initially defined by the distances between the points”, which we now further clarify in the text.
>
> “'the presentation quality of the paper is not good. For example, Tables 1, 2, 3, and 4 are presented but never get referred to in the main paper”
>
> We apologize for this. There was an accidental compiling document in Latex where tables were labeled with section numbers. For E.g. Table 1 was labeled as 4.1 etc. We corrected it in the revised version of the manuscript.
>
>
> Clarity, Quality, Novelty And Reproducibility:
>
> “The developed approach falls short in them given the fact that they merely adopted the existing known techniques including the multi-head attention (MHA) module [1] and triangular multiplicative updates [2] without adding a further major innovation.”
>
> We clearly demonstrated that  DIST achieves state-of-the-art results for instance segmentation of biological structures in electron microscopy data and is competitive with methods operating with similar constraints on ScanNetV2. Furthermore, DIST is a novel solution for point cloud instance segmentation, where using attention and triangulation modules on pairwise point representations has not been done before. We also present a novel framing of instance segmentation as a graph prediction problem, which requires our model to operate on pair representations.
>
>
> “...rational explanation for choosing these components remains untouched”
>
> We have added a paragraph in the method section to further explain why the attention and triangular-multiplication update modules were used.

---

> ### Author Response · Authors · 2022-12-08
> **Further discussion**
>
> Dear Reviewer,
>
> As we are approaching the end of the discussion period, we would appreciate it if you check and reply in this thread, especially since it seems like there is some misunderstanding that we have hopefully clarified in our response and in the revised paper.
> If our response adequately addresses your concerns, please consider raising the score of our submission.
> Thank you very much for your time.
> Authors

---

### Official Review · Reviewer_N2kb · 2022-10-25

**Confidence:** 4
**Correctness:** 4
**Technical Novelty And Significance:** 3
**Empirical Novelty And Significance:** Not applicable
**Recommendation:** 8

**Clarity, Quality, Novelty And Reproducibility:**

The paper is quite clear and easy to understand.
However, there are a number of typos, e.g. "experience microscopist" in page 6 should be "experienced microscopist". Also, the numbers have a weird notation, e.g. "20'000" instead of "20,000".
Some might argue that the method has minimal novelty because it just combines together different techniques from other papers (MHA module from Vasvani et al 2017, triangular update from Jumper et al 2021). However, I see those as building blocks and the proposed method builds on them to obtain a novel solution to a different problem.

**Strength And Weaknesses:**

Strengths:
- The method is quite general and can be used for segmenting both elongated and blob-like structures
- The result is rotation invariant because it is based on distances
- The results are very good, especially for the microscopy images

Weaknesses:
- Related work is missing a discussion of how the proposed method is different from Jumper et al 2021, from which they take some modules.
- The method scales as n^3 with the number n of points. The authors use cropping to reduce the number of nodes processed at one time and then patch the partial results together.

**Summary Of The Paper:**

The paper presents a method for segmenting point clouds by predicting an adjacency graph between the points using a transformer that refines the edge weights of an initial graph. The final segmentation is obtained by graph cuts on the predicted adjacency graph. The method shows good promise in obtaining accurate segmentations for cells in microscopy images.

**Summary Of The Review:**

The paper shows good promise in providing a novel solution to the problem of instance segmentation from point clouds. The paper's strengths of being generic and obtaining good results outweigh its weaknesses of being computationally expensive and missing discussion with some existing work.

---

> ### Author Response · Authors · 2022-11-17
> **Response to reviewer N2kb**
>
> We thank the reviewer for their positive comments and helpful critiques. We have now updated the manuscript to address these concerns, and we respond to the reviewer’s points below.
>
> Weaknesses:
>
> “'Related work is missing a discussion of how the proposed method is different from Jumper et al 2021”
>
> Thank you for pointing this out. We do acknowledge that such a paragraph in related work will improve our paper. We now clarify this in the related work and methods sections.
>
> Briefly, we adapted the triangular multiplication update from Jumper et al. Jumper et al use the triangular update to update their pair representations, which correspond to each pair of columns in their multiple sequence alignment. Their motivation was that this reflected something like triangle inequality. We observed that this also allowed us to define a distance-inspired update for our edge/pairwise point representations. Jumper et al apply these updates sequentially to rows and columns whereas we apply them independently to each and then accumulate those updates. We also include row and column axial attention updates which improved our ability to resolve object instances that were close to each other.
>
> “'The method scales as n3 with the number n of points. The authors use cropping to reduce the number of nodes processed at one time and then patch the partial results together.”
>
> Yes, scaling is an issue with this approach. However, we observe that cropping and stitching barely impact prediction accuracy, because objects are generally continuous and can be accurately stitched across the boundaries given the modest overlap between them.
>
>
>
> Clarity, Quality, Novelty And Reproducibility:
>
> “there are a number of typos”
>
> We have now fixed many of these in the manuscript.

---

### Official Review · Reviewer_zSCn · 2022-10-25

**Confidence:** 3
**Correctness:** 2
**Technical Novelty And Significance:** 2
**Empirical Novelty And Significance:** 2
**Recommendation:** 3

**Clarity, Quality, Novelty And Reproducibility:**

**Clarity**

The writing is clear in general. Some details of algorithm designs and experiment settings need further clarification.

**Quality**

The minus sign in the formula of edge weights is missing. it should be $e_{i,j}=\exp(-0.5 \cdot d^2_{i,j} /s^2)$.

The claim of "*DIST can be applied to other point cloud instance segmentation problems and extends easily to arbitrary dimension point clouds.*" is not well justified, given that the proposed method is not yet able to achieve state-of-the-art performance on ScanNet v2. It is not clear whether the proposed method can indeed provide significant advantages to the problems of point-cloud instance segmentation or even object tracking over time in 3D imaging.

**Novelty**

Applying graph-based representations to the modeling of point cloud data is not new. Using multi-head attention and triangular multiplicative updates to refine the graph Laplacian instead of using transductive propagation might be new.

**Reproducibility**

Pseudocode is included in the appendix.


**Strength And Weaknesses:**


**Weaknesses**

The main weakness of this work is that its problem formulation and experiment setting seem to be different from those adopted in other instance segmentation approaches.  The proposed method actually aims to solve the point-cloud clustering problem.

It is unclear how the class label of each instance is obtained.
In the paragraph of **ground truth generation*, it is described that "*the ScanNetV2 dataset consists of semantically labeled point clouds obtained from LIDAR scanned scenes. In this case, we construct the graph by labeling as $1$ each $P_{i,j}$, if $i$ is in the same instance as $j$ and $0$ if the distance from $j$ to $i$ is less than the average kNN distance between all points. In both cases, the instances are connected subgraphs of the full point cloud graph.*"

So, based on the description above and the procedures mentioned in Section 3, the proposed method only predicts the connected components of the graph as object instances. For the instance segmentation task on ScanNet v2, each object instance should also be assigned the correct semantic label. It is not clear how the results reported in Table 3 are obtained.


**Summary Of The Paper:**

This paper presents a graph-based method to solve point-cloud instance segmentation. The main idea is to train an attention-based deep network that predicts the probability of each pair of points belonging to the same instance. The first step of the proposed method builds a similarity matrix on the point cloud. Each entry of the similarity matrix stores the pairwise edge weight between the corresponding pair of points. The edge weights are derived from Euclidean distances between points plugged into an exponential-decay formula. The second step is built on multi-head attention modules and triangular multiplicative updates to refine the edge weights such that the probabilities of points belonging to the same instances can be better estimated. The final step is simply to obtain connected components by thresholding on the edge probabilities. The proposed method is evaluated on EM data and ScanNet v2.

**Summary Of The Review:**

The proposed method is straightforward. However, the main issue of this work is that its problem formulation and experiment setting do not align with the previous methods.
Although the proposed method is aimed to solve the task of point-cloud instance segmentation, it seems that it only solves point-cloud clustering. It is unclear how the class label of each instance is predicted. The authors need to clarify what is the real problem that the proposed method can address well and choose a more suitable benchmark for evaluation.

---

> ### Author Response · Authors · 2022-11-17
> **Response to reviewer zSCn**
>
> We thank the reviewer for their constructive comments and criticisms. We have revised the manuscript to address these concerns. We hope that these updates and the following response will encourage you to change your opinion about our manuscript.
>
> Weaknesses:
>
> “'... problem formulation and experiment setting seem to be different from those adopted in other instance segmentation approaches...”
>
> Perhaps this is a matter of semantics, but, indeed, we are interested in grouping points into individual object instances and, therefore, identify those instances, but are not interested in semantically classifying them. We refer to this as instance segmentation, because we are segmenting the point cloud into individual instances.
>
> Regardless of semantics, however, we do agree that we adopted different experimental settings in comparison to other available methods for instance segmentation. This framework is tailored towards our primary application to bioimaging where all of the objects are from the same semantic class and we are interested in identifying the individual object instances. We evaluate our method on real cryo-EM and ET datasets and compare our method with the current state-of-the-art algorithms for identifying instances of filaments and membranes in these types of data. Furthermore, the nature of the objects in these datasets is quite different from what is seen in typical computer vision datasets, which makes the problem setting slightly different, but also exciting.
>
> “... the proposed method only predicts the connected components of the graph as object instances.”
>
> Yes, this is exactly true. Our DIST method is designed to predict the graph structure of the point cloud by outputting conditionally independent edge probabilities between each pair of points. Identifying the connected components within the maximum likelihood graph directly solves the instance segmentation problem. It’s not clear why this would be a weakness of our method. On the contrary, it allows us to identify any number of instances and to clearly resolve overlapping and intersecting objects.
>
> “For the instance segmentation task on ScanNet v2, each object instance should also be assigned the correct semantic label.”
>
> The ScanNetV2 semantic instance segmentation task, which was used and highlighted in this manuscript, uses the ground truth (GT) dataset and evaluates the average precision (AP50) denotes the scores with an IoU threshold of 50%. As our DIST model predicts only instances, not semantic labels, class names were obtained post-instance-prediction by comparing the GT with predicted instances and selecting class names with the highest metric value. To ensure no confusion we have now added an expanded explanation in the instance segmentation evaluation paragraph.
>
>
> Clarity:
>
> “Some details of algorithm designs and experiment settings need further clarification.”
>
> ZZe have added additional information clarifying the experimenting setting for the ScanNetV2 dataset. However, regarding the algorithm designs, it would be particularly useful if the reviewer could point us toward the specific parts in the text that were not clarified appropriately.
>
>
> Quality:
>
> “The minus sign in the formula ... missing.”
>
> Thank you for pointing this out. It is now corrected in the manuscript.
>
> “It is not clear whether the proposed method can indeed provide significant advantages to the problems ...”
>
> We have clearly demonstrated that our method dramatically improves the state-of-the-art in point-cloud instance segmentation for identifying filaments and membranes in electron microscopy images.
> However, we disagree with the further assessment. We clearly demonstrate that DIST dramatically improves the state-of-the-art in instance segmentation of biological structures in cryo-EM and ET, improving over the most commonly used software, Amira, and other recently proposed algorithms. We also demonstrated that our method can be applied to other problem settings as we show on Lidar-like datasets (ScanNetV2). Since DIST operates on a distance-based representation of the point cloud, it is naturally invariant to rotation or translation and can work on points in arbitrary dimensions, including object segmentation over time in 2D or 3D.
>
>
> Novelty:
>
> “... is not new.”
>
> To the best of our knowledge, there are no other works that frame instance segmentation directly as a graph prediction problem, where a machine learning model directly outputs a probability distribution over edges between points on the point cloud, and where instances are then predicted from connected components. Methods using graph neural networks operate on k-nearest neighbors-based inputs to the network, but that is not similar to our approach, where we aim to predict a graph as the output of our network. If the reviewer can point us towards literature proposing this strategy before, we would be grateful.

---

> > ### Comment · Reviewer_zSCn · 2022-12-08
> > **ScanNet 3D Semantic Instance Segmentation**
> >
> > Thanks for the detailed response.
> >
> > I would like to know if I misunderstand the experiments on the ScanNet dataset. I have checked the results reported in Table 3 with the "3D Semantic Instance Benchmark" shown on the ScanNet website. It seems that the numbers are directly copied from the benchmark. If that is the case, such comparisons might not provide useful information because, unlike the proposed method that uses ground-truth semantic labels, InsConv and OTOC use their own predicted semantic labels for the evaluation, which is a requirement of the 3D semantic instance task. I think if InsConv and OTOC use the ground-truth semantic labels for evaluation, their AP50 results will be much higher than those shown in the benchmark.

---

> > > ### Author Response · Authors · 2022-12-10
> > > **Clarification for the Semantic Instance segmentation task**
> > >
> > > Dear reviewer,
> > >
> > > Thank you for following up. For the ScanNetV2 benchmark, yes,  we obtained AP50 values directly from the 3D semantic Instance benchmark (scan-net.org). This was the closest comparison available, and still presents an accurate characterization of our ability to identify instances relative to other methods, because semantic and non-semantic instance segmentation results give the same value for AP50 (i.e., AP50 does not depend on the semantic classification).
> > >
> > >  Specifically, the three methods we compare DIST with were trained to output both instance and semantic predictions. However, these methods only use their instance predictions to calculate the AP50 metric. Their semantic predictions are used for the class-level evaluation. Therefore, the AP50 metric is an apples-to-apples comparison between our method and these three methods for instance segmentation.
> > >
> > > We realize this is a confusing distinction and, to reduce confusion, will move the semantic scannet results to the appendix for reference. However, we will only report the 1:1 comparison based on AP50 in the table in the main text. We will also revise the manuscript again to ensure this is clear for the camera-ready version if our paper is accepted.
> > >
> > > We also want to emphasize that our method only considers the spatial location of each point in the point cloud, not additional point features, and yet still achieves competitive results.

---

> > > > ### Comment · Reviewer_zSCn · 2022-12-10
> > > > **ScanNet evaluation metrics**
> > > >
> > > > Thanks for the discussion.
> > > >
> > > > I still need clarification about the evaluation metrics and results reported in the paper and on the ScanNet website.
> > > > I think what we are discussing now is referred to the column of "avg ap 50%" shown in the "3D Semantic Instance Benchmark" on https://kaldir.vc.in.tum.de/scannet_benchmark/semantic_instance_3d?metric=ap50
> > > >
> > > > To my understanding, the methods listed on the benchmark need to predict their AP50 for each class, and the "avg ap 50%" column shows the average of the AP50 values over all classes. Please correct me if I misunderstand the metric. Thank you very much.

---

> > > > > ### Author Response · Authors · 2022-12-11
> > > > > **ScanNet evaluation metric clarification**
> > > > >
> > > > > Dear reviewer,
> > > > > Thank you for your follow back. Yes, we are referring with our comparison to "3D Semantic Instance Benchmark" from link you listed.
> > > > > This benchmark is design to measure the accuracy of instance segmentation prediction with AP50 score. To be more specific, this benchmark measuring 2 things, instance segmentation and semantic-instance segmentation performance. The instance segmentation is measured over all predicted instances as the AP50 score (‘avg ap 50%’ on scan-net.org). Additionally, as you mentioned this benchmar can also be used to evaluated semantic class prediction for each instance. This is presented in the ScanNeV2 table as AP50 score for each semantic class  (‘bathtub, bed, etc.’ on scan-net.org). Therefore as you can see, on the scan-net.org website author of the method disclose both me metrics for instance and semantic-instance. In our paper we want to refer only to instance segmentation measured with ‘avg ap 50%’.
> > > > > We will also revise the manuscript again to make sure this is clear for the camera ready version if our paper is accepted.

---

> > > > > > ### Comment · Reviewer_zSCn · 2022-12-11
> > > > > > **ScanNet 3D evaluation script**
> > > > > >
> > > > > > Thanks for your response, and I am sorry I need to go into the details further.
> > > > > > The following excerpts are from ScanNet's GitHub repo https://github.com/ScanNet/ScanNet/blob/master/BenchmarkScripts/3d_evaluation/evaluate_semantic_instance.py
> > > > > >
> > > > > > ```
> > > > > >     # results: class x overlap
> > > > > >     ap = np.zeros( (len(dist_threshes) , len(CLASS_LABELS) , len(overlaps)) , np.float )
> > > > > > ```
> > > > > >
> > > > > > and
> > > > > >
> > > > > > ```
> > > > > >  avg_dict['all_ap']     = np.nanmean(aps[ d_inf,:,oAllBut25])
> > > > > > ```
> > > > > >
> > > > > > It seems that the APs need to be computed on each class, and then the "all_ap" is obtained by taking the average over all classes. I think this value is different from the one that "is measured over all predicted instances".
> > > > > >
> > > > > > I want to make sure I understand the evaluation metric correctly. Please help clarify the metric and the task. Thank you very much for the discussion.

---

> > > > > > > ### Author Response · Authors · 2022-12-11
> > > > > > > **Classification for scannetv2 evaluation**
> > > > > > >
> > > > > > > Thanks for clarifying, yes, we calculate AP50 over all instances of all semantic classes, ignoring the semantic class labels, in order to measure the ability of our model to identify these instances in a semantic-class independent manner. In contrast, the scannetv2 benchmark calculates max IoU for all instances of each semantic class and then averages these together to get the final mean AP50 over all classes. It is true that this could be lower for semantic instance segmentation methods due to errors in the semantic classification causing an instance that is correctly segmented to be counted as wrong, because it was semantically misclassified. Our benchmark does not penalize these errors.
> > > > > > > This is not a perfect comparison, but it was the best we could do given that we are focusing only on the instance segmentation problem. We want to emphasize that the goal of this comparison is to show that DIST is able to perform instance segmentation reasonably well on natural scene data, generalizing beyond our primary application to instance segmentation of biological structures. However, we have not optimized DIST for this setting and agree that a full application of DIST to these kinds of problems would also require semantic segmentation in the future. We will clarify these points in the manuscript.

---

> ### Author Response · Authors · 2022-12-08
> **Further discussion**
>
> Dear Reviewer,
>
> As we are approaching the end of the discussion period, we would appreciate it if you check and reply in this thread, especially since it seems like there is some misunderstanding that we have hopefully clarified in our response and in the revised paper.
> If our response adequately addresses your concerns, please consider raising the score of our submission.
> Thank you very much for your time.
> Authors

---

### Official Review · Reviewer_FSWQ · 2022-10-26

**Confidence:** 4
**Correctness:** 2
**Technical Novelty And Significance:** 1
**Empirical Novelty And Significance:** 2
**Recommendation:** 3

**Clarity, Quality, Novelty And Reproducibility:**

As discussed in the previous section, I think that the quality of this work is low with a reduced impact in the ICLR research community and a limited novelty.

**Strength And Weaknesses:**

I have several concerns regarding this paper. First of all, it seems to me that this paper is focused on a specific applications (i.e., segmentation of biological structures in electron microscopy), which might be of limited interest for the ICLR audience. In addition, I think that the theoretical contribution of the proposed method is limited, it doesn't seem to me that the proposed network is very innovative, since it is a quite standard network with multi head attention. I have also some concerns on the experiments validation of the proposed method: in the performance comparison the authors do not consider the most well-known state-of-the-art methods for point cloud segmentation (in particular in the experiment on ScanNetV2 in Table 2). For this reason, it is not possible to assess the performance of the proposed method.

Minor comment:
the authors do not used the correct template for ICLR.

**Summary Of The Paper:**

This paper presents a method for point cloud segmentation specifically focused on biological applications. The proposed method defines a representation of the input point cloud using the pairwise distances between points. This representation is fed into deep neural network with attention layers in order to obtain the predicted instance segmentation.

**Summary Of The Review:**

This paper presents an application of deep learning to a very specific application field (i.e., segmentation of biological structures in electron microscopy). I believe that this application is of limited interest for the ICLR community and the technical novelty of the proposed method is very limited. For these reasons, I don't think that ICLR is the correct venue for this work.

---

> ### Author Response · Authors · 2022-11-17
> **Response to Reviewer FSWQ**
>
> Thank you for your comments and suggestions. We respond to your specific concerns below.
>
>
> Strength And Weaknesses:
>
> “'First of all, it seems to me that this paper is focused on a specific applications...which might be of limited interest for the ICLR audience”
>
> We do focus on the important problem of segmenting biological structures in electron microscopy data as one application of our DIST method, but the method is general purpose and can be applied to other problems as we demonstrated in our experiments on ScanNetV2.
>
> Moreover, we believe that applications to bioimaging will be of interest to the ICLR community. Not only are these imaging modalities and methods for processing data from cryoEM and cryo-ET critical for advancing the general understanding of biology, but these datasets also present new opportunities and new problems for the ML community to work on. The ICLR community is clearly interested in these sorts of applications with broad interest and many papers are being presented on applications to medical imaging, protein and gene expression analysis, protein structures, etc. The ICLR paper call, itself, clearly states that “...applications in audio, speech, robotics, neuroscience,  biology or any other field” are appropriate contributions for this conference.
>
>
> “'I think that the theoretical contribution of the proposed method is limited, it doesn't seem to me that the proposed network is very innovative.”
>
> We use a point cloud transformer, which is invariant to the specific orientation of the point cloud, and a novel graph prediction output with a graph-cut-based objective for instance segmentation. This allows us to identify an arbitrary number of instances and have invariance to rotations and translations of the point clouds. Our point cloud transformer leverages pair representations of the points and axial updates, both of which are new innovations in the transformer field. To our knowledge, this model architecture and problem formulation is novel for instance segmentation of point clouds. If the reviewer can point us towards literature proposing this strategy before, we would be grateful.
>
>
> “'I have also some concerns on the experiments validation...the authors do not consider the most well-known state-of-the-art methods...”
>
> As explained in the related work section, we selected two methods that most closely matched the problem setting of DIST. We do not dispute that DIST has not demonstrated state-of-the-art performance on ScanNetV2, and never claimed otherwise. We compared primarily against InsConv, SegGroup and OTOC, as these methods (similarly to DIST) used only point cloud coordinates (without RGB features) for prediction or predict graph representations of a point cloud, similarly to DIST. However, to allow the reviewer and the reader to assess the performance of our model with the most well-known state-of-the-art methods for point cloud instance segmentation, we expanded Appendix B by adding a supplement table with these metrics (Now Appendix G). Previous Appendix B included every model that took part in the ScanNetV2 instance segmentation challenge with their mAP50 score. We expanded this by adding Appendix G which now includes a detailed metric for each object class for 3 state-of-the-art models (Mask3D, SPFormer and SoftGroup++).
>
> Minor comment:
>
> “The authors do not used the correct template for ICLR”
>
> The missing header has now been corrected.

---

> ### Author Response · Authors · 2022-12-08
> **Further discussion**
>
> Dear Reviewer,
>
> As we are approaching the end of the discussion period, we would appreciate it if you check and reply in this thread, especially since it seems like there is some misunderstanding that we have hopefully clarified in our response and in the revised paper.
> If our response adequately addresses your concerns, please consider raising the score of our submission.
> Thank you very much for your time.
> Authors

---

> > ### Comment · Reviewer_FSWQ · 2022-12-13
> > **Response to authors**
> >
> > I thank the authors for their detailed response. However, I am still not convinced that the problem tackled in this paper is of broad interest for the ICLR community especially because the technical novelty seems limited. For this reason, I will keep my score.

---

### Decision · Program_Chairs · 2023-01-20

**Decision:**

Reject

**Justification For Why Not Higher Score:**

There are major limitations of the work in whether the work is of broad interest, lacking comparison with latest point cloud segmentation networks, and exposition. Given its current status, the meta-reviewer does not recommend to accept the submission.

**Justification For Why Not Lower Score:**

N/A

**Metareview: Summary, Strengths And Weaknesses:**

The paper proposes a method for point cloud segmentation on biologic data leveraging pairwise distances between points and multi-head attention. The performance in the main experiments seems good, though some reviewers feel the comparison with existing work is not thorough. The major weakness of the paper includes marginal novelty, missing comparison and discussion to latest literatures, and some issues in writing.